# Performance and Ruminal Parameters of Boer Crossbred Goats Fed Diets that Contain Crude Glycerin

**DOI:** 10.3390/ani9110967

**Published:** 2019-11-13

**Authors:** Higor Bezerra, Edson Santos, Juliana Oliveira, Gleidson Carvalho, Fabiano Silva, Meiry Cassuce, Alexandre Perazzo, Anderson Zanine, Ricardo Pinho

**Affiliations:** 1Department of Animal Science, Federal University of Vale do São Francisco, Petrolina, Pernambuco 56304-917, Brazil; higorfabiozoo@hotmail.com; 2Department of Animal Science, Federal University of Paraíba, Areia, Paraíba 58397-000, Brazil; edsonzootecnista@yahoo.com.br (E.S.); oliveirajs@yahoo.com.br (J.O.); alexandreperazzo@hotmail.com (A.P.); 3Department of Animal Science, Federal University of Bahia, Salvador, Bahia 40210-730, Brazil; gleidsongiordano@yahoo.com.br; 4Department of Animal Science, State University of Southwest of Bahia, Itapetinga, Bahia 45700-000, Brazil; ffsilva@pq.cnpq.br; 5Department of Animal Science, Federal Rural University of Pernambuco, Recife, Pernambuco 50670-901, Brazil; meirycassuce@gmail.com; 6Department of Animal Science, Federal University of Maranhão, Chapadinha, Maranhão 65500-000, Brazil; ricardo-zootec@hotmail.com

**Keywords:** alternative feed, biodiesel, feedlot, glycerol, weight gain

## Abstract

**Simple Summary:**

There is an increasing interest in the use of by-products, such as crude glycerin in animal feeding, and many studies emphasize that the addition of these ingredients can reduce feed costs, avoid environmental problems, and serve as good energy sources for ruminants. However, there is little information available on the replacement of ground corn with crude glycerin in goat diets. We analyzed the effects of replacing corn with crude glycerin up to 150 g/kg of diet on feed intake and growth performance of goats. This substitution reduced consumption, digestibility, and performance.

**Abstract:**

This study aimed to evaluate the effects of four levels of crude glycerin (0, 50, 100, or 150 g/kg on a dry matter basis) on intake, digestibility, production performance, and ruminal parameters for finishing Boer crossbred goats. Thirty-two crossbred, castrated Boer × undefined breed goat kids, with an initial average weight of 17.8 ± 2.2 kg and approximately four months old, were distributed in a completely randomized design, with four treatments and eight repetitions. The dry matter and neutral detergent fiber intakes, both in g/day and percent of body weight, linearly decreased (*p* ≤ 0.05) with increased inclusion levels of crude glycerin in the diet. The dietary crude glycerin levels linearly decreased (*p* ≤ 0.01) the digestibility coefficients of ether extract and quadratically increased (*p* = 0.04) digestibility coefficients of neutral detergent fiber. The final weight, total weight gain, and average daily gain for the animals showed a linear decrease (*p* ≤ 0.02) as dietary crude glycerin levels increased. The addition of crude glycerin caused a linear increase in ruminal pH (*p* ≤ 0.01), which ranged from 6.27 to 6.49 for diets with 0 and 150 g/kg crude glycerin, respectively. The concentration of ruminal NH_3_–N exhibited a linear decrease as the crude glycerin inclusion levels increased (*p* ≤ 0.01). Total short-chain fatty acid (SCFA) concentration, individual molar ratio, and the acetate: Propionate ratio in the ruminal fluid of the animals were not influenced (*p* ≥ 0.07) by the dietary crude glycerin levels. These data indicate that crude glycerin should not be used to replace ground corn in the diets of growing goats that are finished in a feedlot because the substitution reduces the intake and digestibility of several nutrients and decreases performance.

## 1. Introduction

The growing production of biodiesel requires viable alternatives for the by-products generated during the procurement process. The main by-product of the biodiesel industry is crude glycerin, which represents approximately 10% of the biodiesel produced [1]. Crude glycerin has become a potential alternative feed ingredient for livestock in many species including poultry [2], swine [3], cattle [4,5] and small ruminants [6,7]. 

In ruminants, different quantities of glycerin are converted to volatile fatty acids, particularly to butyrate and propionate, which is the main precursor for gluconeogenesis in the liver [7,8], and can provide energy for cellular metabolism [4]. From a glucogenic perspective, the inclusion of crude glycerin will increase dietary glucogenic potential when glycerin replaces corn in goat diets [7]. Although glycerol may represent an alternative energy source for livestock, there are unanswered questions regarding the handling, inclusion rates, impact, and feeding value in ruminant diets [9]. To our knowledge, there is also little information available on replacement of ground corn with crude glycerin in goat diets. Thus, it is important to evaluate the effects of crude-glycerin-supplemented diets on the performance of goats. We hypothesized that crude glycerin can partially replace the ground corn in diets for Boer crossbred goat kids. Therefore, the objective of the current study was to determine the effects of dietary addition of crude glycerin on intake, digestibility, performance, and ruminal parameters of Boer crossbred goat kids.

## 2. Materials and Methods 

All animal management and care procedures were in accordance with the guidelines approved by the Federal University of Bahia Animal Use and Care Committee (n. 08/2013).

### 2.1. Location, Animals, and Diets

The experiment was conducted at the Experimental Farm of the School of Veterinary Medicine and Animal Science at the Federal University of Bahia (UFBA) in São Gonçalo dos Campos, Bahia, Brazil, between November 2013 and January 2014. 

Thirty-two approximately 4-month-old castrated Boer × undefined breed goat kids, with an average initial body weight (BW) of 17.8 ± 2.2 kg, were tested. These animals were distributed in a completely randomized design, with four treatments and eight repetitions per treatments. Four levels of crude glycerin (0, 50, 100, or 150 g/kg) based on dietary dry matter (DM; Table 1) were used.

The animals were housed in individual 2-m^2^ pens in covered sheds that were equipped with feeders and water throughout the trial period. The experiment lasted for 69 days: 15 days of adaptation of the animals to the facilities and diets and 54 days of data collection. All the animals were identified and treated with ivermectin (Merial, SP, Brazil), for the control of parasites, before the start of the experiment.

The forage:concentrate ratio of diets was 60:40, and animals were fed with total mixed ration at 08:00 and 16:00. Sorghum silage (*Sorghum bicolor* (L). Moench) was used as roughage (Table 2). The leftovers were weighed daily, and the amount of feed supplied was adjusted to allow for leftovers of up to 15% of the amount supplied. Water was supplied ad libitum.

The diets were formulated to be isonitrogenous and meet the nutritional requirements of growing goats, with an average daily gain of 150 g, according to a previous report [12].

### 2.2. Feed Intake, Nutrient Digestibility and Animal Performance

Individual intake was assessed by subtracting the refusals from the amount of diet offered to each animal. Dry matter (DM), organic matter (OM), crude protein (CP), ether extract (EE), neutral detergent fibre (NDF), non-fibrous carbohydrate (NFC), and total digestible nutrient (TDN) intakes were assessed and expressed in g/animal/day (g/day). Values relative to DM and NDF intake were also expressed in per cent of body weight (BW).

The digestibility assay was conducted with the 32 goats between days 38 and 45 of confinement using total fecal collection. The first 3 days were used to adapt the animals to the collection bags, followed by 5 days of total fecal collection [13]. After the total fecal production of each animal was recorded, aliquots of approximately 100 g/kg of the total collection were transferred to individual labelled plastic bags and stored in a freezer.

During the digestibility assay, samples of the supplied feed were collected and submitted to pre-drying in a forced ventilation oven at 55 °C for 72 h. Next, samples were ground using a Wiley mill with a 1-mm sieve, prepared individually according to the animal and duly packed in labelled plastic containers for subsequent laboratory analysis. DM, OM, CP, EE, NDF, and NFC digestibility coefficients were calculated from the following previously proposed equation [10]:

DC = ([kg of fraction ingested − kg of fraction excreted]/[kg of fraction ingested]) × 100.

Refusals were collected daily at 07:00 before the morning feed was delivered and weighed in a digital scale to determine DMI. DMI was obtained by adjusting the amount of feed offered to the goats to allow for 5–15% refusal.

All goats were weighed before the morning feeding at the beginning of the experiment and every 14 days at the same time of day and before transportation to the slaughterhouse to obtain their total weight gain (TWG). The average daily gain (ADG) was determined by dividing BW gain (initial full BW − final full BW) by the number of days in the study. Feed conversion (FC) was calculated as the ratio between kg DMI/kg BW gain. Feed efficiency (FE) was calculated as the ratio between kg BW gain/kg DMI.

### 2.3. Chemical Analysis

During the experimental period, samples of the diets offered and refusals were collected weekly, packed in labelled plastic bags, and stored in a freezer at −20 °C. After thawing, the samples of the ingredients and refusals were pre-dried in a forced air oven at 55 °C for 72 h. Next, the samples were ground in a Wiley mill with a 1-mm sieve, packed in lidded plastic containers, labelled, and prepared for laboratory analysis.

Samples of the ingredients, diets, and refusals for each experimental unit (represented by the animal) and for each experimental period were frozen for further analysis. DM (method 934.01; [14]), ash (method 930.05; [14]), CP (method 920.87; [14]), and EE (method 920.85; [14]) compositions were determined from the feeds. The OM was calculated by the difference between DM and ash contents. Determination of the acid detergent fiber (ADF) was obtained following a published method [15], and NDF was determined according to the method described by [16]. Refusals and feces were analyzed for DM, OM, CP, EE, ash, and NDF, all of which were used to calculate the TDNs. 

The total glycerol and methanol content of crude glycerin were analyzed by gas chromatography (TRACE GC Ultra, Thermo Electron Corporation, Rodano, Italy), using a 30 m × 0.25 mm × 0.25 μm capillary column (Tracsil TR-FFAP; Teknokroma, Barcelona, Spain) equipped with a flame ionization detector, as previously described [17].

### 2.4. Calculated Composition

NFC was estimated using the following previously described equation [10]: NFC = 100 − (%CP + %EE + %Ash + %NDF). 

The TDN was calculated according to [11], with the equation:

TDN (%) = DCP + DNFC + (DEE × 2.25) + DNDF, where DCP is the digestible CP, DEE is the digestible EE, DNDF is the digestible NDF, and DNFC is digestible NFC.

### 2.5. Sampling and Analysis of Ruminal Fluid

A trial following a 4 × 4 Latin square design was carried out with eight fistulated Boer crossbred goats. The animals had a BW of 40 ± 2.5 kg with a body score of 3 (on a scale of 1 to 5). These goats were confined in individual stalls with wooden slat floors. The animals were distributed in two groups of 4 animals each. The treatments corresponded to four levels of crude glycerin (0, 50, 100, or 150 g/kg) based on dietary DM (Table 1).

The trial comprised four experimental periods, each with a 15-day of duration. The first 10 days were for adaptation, and the remaining 5 days were for samples collection. Goats with ad libitum access to feed and water were fed at 8:00 and 16:00 to allow for refusals of 15%.

On day 14 of each experimental period, rumen content samples were collected to determine pH, ammonia nitrogen (NH_3_–N), and short-chain fatty acid (SCFA) concentrations at 0, 4, 8, 12, 16, 20, and 24 h. Manual collection of ruminal samples coincided with the time before the diet was provided at 08:00. Approximately 100 g of rumen content was taken from three places in the rumen, strained through four layers of cheesecloth, and preserved in individual plastic tubes for each goat. Immediately after collection, samples were evaluated with a pH meter (HANNA, model HI 96108, Tamboré Barueri, Brazil) before storage at −20 °C. For SCFA analysis, the material was thawed and centrifuged (HETTICH, model Mikro 200, Tuttlingen, BW, Germany) at 5200× *g* for 10 min. The supernatant was removed, and 0.5 mL was transferred to a 1.0-mL Eppendorf tube with 0.5 mL 25% metaphosphoric acid solution. SCFA were measured using high-performance liquid chromatography (HPLC; SPD-10 AVP, Shimadzu Corporation, Osaka, Japan) coupled to an ultraviolet (UV) detector at 210 nm [18]. The remainder of the supernatant was transferred to another Eppendorf tube for NH_3_–N analysis using a previously described colorimetric method [19]. 

### 2.6. Statistical Analysis

Data from feed intake, nutrient digestibility, and animal performance were subjected to analysis of variance (ANOVA) in a completely randomized design with four treatments, namely 0, 50, 100, or 150 g/kg inclusion of crude glycerin, and eight replications. The initial weight of the goats was considered as a covariate in the statistical model. Data from ruminal parameters were analyzed in a 4 × 4 Latin square.

The results were interpreted through decomposition of the orthogonal polynomials in linear and quadratic models using the PROC MIXED function of the SAS software (version 9.1, SAS Institute Inc., Cary, NC, USA). Homogeneity of variance between treatments was assumed, and the degrees of freedom were estimated using the Kenward–Roger method. All statistical procedures were performed using a value of 0.05 as the critical level of probability for a type I error. 

## 3. Results

### 3.1. Nutrient Intake

The DM and NDF intakes linearly decreased (*p* ≤ 0.05) as the crude glycerin inclusion level in the diet increased (Table 3). Additionally, there was a linear reduction in OM, CP, EE, NFC, and TDN intakes (*p* ≤ 0.05) due to increased levels of crude glycerin in the diet (Table 3).

### 3.2. Digestibility Coefficients

Apparent digestibility of DM, OM, CP, and NFC were not affected by the dietary crude glycerin levels (*p* ≥ 0.07; Table 4). However, the dietary crude glycerin levels linearly decreased (*p* ≤ 0.01) the digestibility coefficients of EE and quadratically increased (*p* = 0.04) the digestibility coefficients of NDF, with maximum digestibility estimated at 60.54% with 47.80 g/kg of crude glycerin. 

### 3.3. Performance

The final weight (FW), TWG, and ADG of the animals showed a linear decrease (*p* ≤ 0.02) as dietary crude glycerin levels increased (Table 5). The different levels of crude glycerin did not affect (*p* ≥ 0.19) FC or FE, which showed means of 6.58 (kg/kg) and 0.163 (kg/kg), respectively. 

### 3.4. Rumen Fermentation

The addition of crude glycerin linearly increased ruminal pH (*p* ≤ 0.01). pH ranged from 6.27 to 6.49 for diets with 0 and 150 g/kg crude glycerin, respectively (Table 6).

The concentration of ruminal NH_3_–N linearly declined as the crude glycerin inclusion levels increased (*p* ≤ 0.01). The 150 g/kg crude glycerin (19.02 mg/dL) treatment produced the lowest concentrations, whereas 0 g/kg (26.17 mg/dL) showed the highest means (Table 6). 

Total SCFA concentration, individual molar ratio, and the acetate:propionate (A:P) ratio in the ruminal fluid of the animals were not influenced (*p* ≥ 0.07) by the dietary crude glycerin levels (Table 6). 

## 4. Discussion

In this study, the DMI ranged from 826 g/day (control diet) to 608 g/day (diet with 150 g/kg crude glycerin). These data demonstrate the rejection of diets with increasing levels of glycerin due to specific characteristic of the glycerin itself, perhaps the methanol content. These results indicate that elevated crude glycerin and, consequently, the increased methanol content of the diet, may cause feed rejection and a resulting decrease in DMI in small ruminants, as previously reported [20]. Thus, the potential cause of intake restriction among small ruminants may be methanol—and not glycerol—the latter of which is easily absorbed directly by the ruminal epithelium, metabolized in the liver, and directed to gluconeogenesis via the action of the enzyme glycerol kinase, which converts it into glucose [21]. The higher infusion of methanol in the rumen can increase methane production and thus decrease the utilization of the diet due to energy loss in methane production [22].

The goats showed a greater decrease in EE intake compared to the intake of other nutrients (Table 3). DM exhibited a decrease of approximately 219 g/d with the inclusion of 150 g/kg crude glycerin compared with the treatment without the inclusion of glycerin, whereas EE showed a decrease of approximately 12.7 g/d with the addition of 150 g/kg crude glycerin relative to the control treatment. This effect may be explained by the decreased EE content in the diet with crude glycerin (Table 1). Consequently, it was also less consumed, given the lower EE availability.

In addition to methanol, other factors may limit the intake of diets with crude glycerin, especially salts and residues contained in recycled oils and reagents used in the transesterification process [6,7,17]. One study reported that the inclusion of up to 400 g/kg glycerin (which contained 53.8 g/kg sodium) reduced the intake of diets with glycerin by sheep and attributed this decrease to the high sodium content in the diet [7].

The similar nutrient contents of the diets might explain the similar digestibility of these nutrients between diets and why the inclusion of crude glycerin did not promote changes in diet digestibility.

The lowest NDF digestibility, which appeared in the highest dose of crude glycerin, may be due to the reduction of certain microbial groups, including *Butyrivibrio fibrisolvens* (fibrolytic) bacteria. One study reported a 53% decrease in *B. fibrisolvens* DNA concentration when 108 g of glycerol per kg DM was added to the diet (compared to a diet without glycerol) [23].

The reduced ADG and TWG of goats fed glycerin in this study were likely a direct result of decreased DMI, as discussed previously. Reduced DMI affiliated with increased dietary glycerin content may be a result of high methanol content in the diets [20]. We emphasize that although crude glycerin did not affect FE and FC, the decrease in ADG limits the substitution of corn with glycerin in order to minimize feed cost.

The pH values reported in this study corroborate a previous study that reported the optimum pH range for the healthy development of ruminal microbial activity is between 6.2 and 7.2 [24]. These results may be due to the decreased starch supply from the inclusion of crude glycerin because the consumption of highly fermentable diets leads to a more acidic ruminal pH compared to that physiologically observed in animals with low energy intake. This effect is true for starch, which is highly fermentable due to amylolytic bacteria. Indeed, its reduction in the diets is correlated with the reduction in this population of microorganisms and consequent ruminal acidification. Crude glycerin contains mostly glycerol, which is primarily absorbed directly by the ruminal epithelium, metabolized in the liver, and directed to gluconeogenesis in ruminants [21].

The decreased ammonia concentration may result from the lower CP intake. This possibility is supported by the decreased CP amount in the rumen as dietary CP intake increases [25,26]. All NH_3_–N results found in this study exceeded 5 mg/dL, which is the minimum level of ruminal NH_3_–N required for preserving normal rumen functions [27]. These NH_3_–N values corroborate a previous study [28], and these data suggest that the maximum fermentative activity occurs at rumen NH_3_–N concentrations of 19–23 mg/dL of ruminal liquid. 

## 5. Conclusions

Crude glycerin should not be used to replace ground corn in the diets of growing goats being finished in a feedlot because the substitution reduces the intake and digestibility of several nutritional fractions and decreases performance.

## Figures and Tables

**Table 1 animals-09-00967-t001:** Ingredients and chemical composition of experimental diets.

Dietary Ingredient (%)	Dietary Crude Glycerin Levels (g/kg Dry Matter)			
	0	50	100	150
Diet composition				
Cornmeal	18.0	12.0	6.00	0.00
Soybean meal	20.5	21.5	22.5	23.5
Crude glycerin	0.0	5.00	10.0	15.0
Mineral supplement	1.50	1.50	1.50	1.50
Sorghum silage	60.0	60.0	60.0	60.0
Chemical composition, %				
DM	55.5	55.7	56.0	56.2
Organic matter [OM] ^1^	94.1	93.7	94.1	94.2
Mineral matter [MM] ^1^	5.08	5.23	5.39	5.54
Crude protein [CP] ^1^	14.9	15.0	15.1	15.1
Ether extract [EE] ^1^	3.13	2.84	2.55	2.26
Neutral detergent fiber [NDF] ^1^	34.9	34.3	33.7	33.1
Acid detergent fiber [ADF] ^1^	16.7	16.6	16.6	16.6
Methanol ^1^	0.00	0.33	0.66	0.99
Calculated composition, %				
Non-fibrous carbohydrate [NFC] ^1,2^	41.9	42.6	43.3	44.0
Total digestible nutrients [TDN] ^3^	64.0	64.0	64.0	64.0

^1^ Value expressed in per cent dry matter; ^2^ Estimated by the equation NFC = 100 − (%CP + %EE + %Ash + %NDF) by [10]; ^3^ Estimated by the equation TDN (%) = DCP + DNFC + (DEE × 2.25) + DNDF by [11].

**Table 2 animals-09-00967-t002:** Chemical composition of ingredients in experimental diets.

Item (%)	Ingredient			
	Sorghum Silage	Cornmeal	Soybean Meal	Crude Glycerin
Dry matter	33.6	88.6	87.3	94.0
Organic matter ^1^	96.7	98.5	93.5	96.4
Mineral matter ^1^	3.29	1.54	6.48	3.60
Crude protein ^1^	7.55	6.42	45.03	ND
Ether extract ^1^	3.05	5.15	1.84	ND
Neutral detergent fibre ^1^	49.0	13.1	15.5	ND
Acid detergent fibre ^1^	26.2	1.30	3.63	ND
Glycerol	0.00	0.00	0.00	43.4
Methanol	0.00	0.00	0.00	6.6
Calculated composition				
Non-fibrous carbohydrate ^1,2^	37.1	73.8	31.2	96.4
Total digestible nutrients ^1,3^	55.0	81.1	80.1	ND

ND = not determined; ^1^ Value expressed in % of dry matter; ^2^ Estimated by the equation NFC = 100 − (%CP + %EE + %Ash + %NDF) by [10]; ^3^ Estimated by the equation TDN (%) = DCP + DNFC + (DEE × 2.25) + DNDF by [11].

**Table 3 animals-09-00967-t003:** Intakes of dry matter (DM), organic matter (OM), crude protein (CP), ether extract (EE), neutral detergent fiber (NDF), non-fibrous carbohydrates (NFC), and total digestible nutrients (TDN) by goats fed diets with crude glycerin.

Item	Dietary Glycerin Level (g/kg)				SEM ^1^	*p*-Value	
	0	50	100	150		L ²	Q ³
Intake (g/day)							
DM	827	733	714	608	27.4	0.01	0.89
OM	793	704	683	581	26.3	0.01	0.87
CP	110	105	95.7	85.5	3.56	0.01	0.67
EE	21.6	16.6	14.0	8.92	0.999	<0.01	0.96
NDF	299	278	272	202	11.5	0.01	0.16
NFC	362	313	301	284	10.1	0.01	0.57
TDN	588	512	492	397	19.6	0.01	0.68
Intake (% body weight [BW])							
DM	3.16	2.92	2.87	2.63	16.78	0.01	0.72
NDF	1.14	1.11	1.09	0.87	17.76	0.01	0.09

^1^ SEM = Standard error of the mean; L ² = Linear effect; Q ³ = Quadratic effect. Abbreviations—DM = dry matter; OM = organic matter; CP = crude protein; EE = ether extract; NDF = neutral detergent fiber; NFC = non-fibrous carbohydrate, TDN = total digestible nutrients.

**Table 4 animals-09-00967-t004:** Nutrient apparent digestibility and total digestible nutrients (TDN) of goats fed diets with crude glycerin.

Item	Dietary Glycerin Level (g/kg)				SEM ^1^	*p*-Value	
	0	50	100	150		L ²	Q ³
**Digestibility (%)**							
DM	69.8	69.0	68.2	65.1	0.90	0.07	0.37
OM	71.3	70.7	70.1	66.2	0.86	0.08	0.34
CP	67.1	70.3	69.0	67.2	0.85	0.31	0.09
EE	76.5	62.9	56.3	38.3	3.26	<0.01	0.31
NDF	58.6	60.4	58.4	51.6	1.39	0.02	0.04
NFC	82.8	80.5	81.6	81.4	0.57	0.22	0.54

^1^ SEM = Standard error of the mean; L ² = Linear effect; Q ³ = Quadratic effect. Abbreviations—DM = dry matter; OM = organic matter; CP = crude protein; EE = ether extract; NDF = neutral detergent fibre; NFC = non-fibrous carbohydrate.

**Table 5 animals-09-00967-t005:** Average initial weight (IW), final weight (FW), total weight gain (TWG), average daily gain (ADG), feed conversion (FC, kg dry matter intake [DMI]/kg gain), and feed efficiency (FE, kg gain/kg DMI) of goats fed diets with crude glycerin.

Item	Dietary Glycerin Level (g/kg)				SEM ^1^	*p*-Value	
	0	50	100	150		L ^2^	Q ^3^
IW (kg)	17.8	17.9	17.9	17.6	-	-	-
FW (kg)	26.2	25.2	24.9	23.1	0.59	0.02	0.09
TWG (kg)	8.42	7.24	6.98	5.33	0.339	0.01	0.12
ADG (g/day)	122	105	101	87.1	0.01	0.01	0.23
FC	6.30	6.36	6.84	6.83	0.3495	0.53	0.38
FE	0.166	0.165	0.160	0.160	0.0078	0.69	0.19

^1^ SEM = Standard error of the mean; L ² = Linear effect; Q ³ = Quadratic effect.

**Table 6 animals-09-00967-t006:** Rumen fermentation parameters of goats fed diets with crude glycerin.

Item	Crude Glycerin Inclusion Level (g/kg)				SEM ^1^	*p*-Value	
	0	50	100	150		L ²	Q ³
Ruminal pH	6.27	6.34	6.42	6.49	0.028	<0.01	0.96
NH_3_–N (mg/dL)	26.2	23.1	20.7	19.0	1.21	<0.01	0.15
Acetate (mol/100 mol)	67.0	64.3	64.4	64.1	0.54	0.08	0.12
Propionate (mol/100 mol)	25.8	28.2	28.3	28.4	0.49	0.07	0.11
Butyrate (mol/100 mol)	7.24	7.44	7.38	7.49	0.094	0.38	0.76
Total SCFA (mol/L)	146	142	143	141	1.85	0.24	0.72
Acetate:propionate (A:P)	2.74	2.35	2.35	2.34	0.066	0.07	0.09

^1^ SEM = Standard error of the mean; L ² = Linear effect; Q ³ = Quadratic effect.

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
