# Peer review of "Performance and Ruminal Parameters of Boer Crossbred Goats Fed Diets that Contain Crude Glycerin"

_animals, 2019, doi:10.3390/ani9110967_

Round 1
Reviewer 1 Report
The paper investigates effects of increasing inclusion levels of crude glycerin in goat diets on performance and ruminal characteristics. The paper is well written, the study design is sound, the parameters analyzed are sufficient to answer the question posed. I recommend to publish this paper in Animals and have only one minor comment, see below.
Line 266 etc. How high was Na content in glycerin used in the present study? Is any other information available on further residues in glycerin?
Author Response
Zarol Han
Academic Editor
Animals
We are pleased our paper was accepted and appreciate the attention and contributions the article received from your team of reviewers. All reviewer comments have been addressed, as indicated below and in the attached file. We carefully observed the style and form of Animals and revised the manuscript accordingly. The responses to the questions/comments by the reviewers are provided below, and all the changes in the manuscript are highlighted in yellow.
Sincerely yours,
Anderson de Moura Zanine
|
Comments for the author |
Correction and answers |
|
Discussion |
|
|
Line 266 etc. How high was Na content in glycerin used in the present study? Is any other information available on further residues in glycerin? |
We have not analysed the mineral content of diets. However, the literature reports that the glycerin levels used in the diets were not possibly limited by excess sodium. |
Reviewer 2 Report
Line 51-52: There are many data about performances or ruminants fed diet with glycerine. For example see "Semina: Ciências Agrárias, Londrina, v. 36, n. 3, maio/jun. 2015". The present study is similar to: Favaro et al. (2015) - Glycerin in cattle feed: intake, digestibility, and ruminal and blood parameters (see abstracts).
Line 52: check English
Line 61: check P. Chanjula (2017) - Use of Crude Glycerin as an Energy Source for Goat Diets: A Review. Dairy and Vet. Sci. J, 2 (1)
Table 1 and table 2: add acronyms (OM, CP, EE, NDF, Alf, NFC, TDN) in the table
Line 134: 38th and 45th day
Line 136: ...followed by 5 day of total fecal collection [10]
Line 146: the chemical composition of refusals could help to better understand the results; please add if available
Line 167: TDNs were calculated
Line 177: check English; check DNDF
Line 181: ....mean body weight (BW)...
Line 181: delete "approximately", add StdDev of BW
Line 185: The trial consisted....
Line 191: describe the rumen fluid sampling method
Table 6: use "rumen" instead of "ruminal"
Line 253 - 255: there are secondary effects of methanol in animal nutrition and digestibility of feeds; please add some information about it and describe observed secondary effect (if any) on goats
Line 269: Did you analyze the minerals of diets (Na for example)?
Line 274 - 275: check English, or rewrite, and explain
Line 278: see previous note about methanol in the diet
Line 290-292: not clear; rewrite
Author Response
Zarol Han
Academic Editor
Animals
We are pleased our paper was accepted and appreciate the attention and contributions the article received from your team of reviewers. All reviewer comments have been addressed, as indicated below and in the attached file. We carefully observed the style and form of Animals and revised the manuscript accordingly. The responses to the questions/comments by the reviewers are provided below, and all the changes in the manuscript are highlighted in yellow.
Sincerely yours,
Anderson de Moura Zanine
|
Comments for the author |
Correction and answers |
|
Introduction |
|
|
Line 51-52: There are many data about performances or ruminants fed diet with glycerine. For example see "Semina: Ciências Agrárias, Londrina, v. 36, n. 3, maio/jun. 2015". The present study is similar to: Favaro et al. (2015) - Glycerin in cattle feed: intake, digestibility, and ruminal and blood parameters (see abstracts). |
We have agreed. Some articles present the use of crude glycerin in the ruminant diet. However, few report the use of glycerin in substitution of corn. |
|
Line 52: check English |
We have checked the English style. |
|
Line 61: check P. Chanjula (2017) - Use of Crude Glycerin as an Energy Source for Goat Diets: A Review. Dairy and Vet. Sci. J, 2 (1) |
Although there are several articles evaluating the effect of inclusion of crude glycerin as a source of energy, few studies have evaluated the replacement of corn with crude glycerin. |
|
Material and methods |
|
|
Table 1 and table 2: add acronyms (OM, CP, EE, NDF, Alf, NFC, TDN) in the table |
We have added the acronyms. |
|
Line 134: 38th and 45th day |
We have corrected. |
|
Line 136: ...followed by 5 day of total fecal collection [10] |
We have corrected. |
|
Line 146: the chemical composition of refusals could help to better understand the results; please add if available |
We have the chemical composition of the refusals in order to calculate the intake and nutrient apparent digestibility but it is not common in the manuscripts to have a table with the chemical composition of the refusals. |
|
Line 167: TDNs were calculated |
We have added the sentence. |
|
Line 177: check English; check DNDF |
We have corrected. |
|
Line 181: ....mean body weight (BW)... |
We have added the sentence. |
|
Line 181: delete "approximately", add StdDev of BW |
We have corrected. |
|
Line 185: The trial consisted... |
We have corrected. |
|
Line 191: describe the rumen fluid sampling method |
We have described more detailed the rumen fluid sampling method. |
|
Results |
|
|
Table 6: use "rumen" instead of "ruminal" |
We have replaced ruminal with rumen |
|
Discussion |
|
|
Line 253 - 255: there are secondary effects of methanol in animal nutrition and digestibility of feeds; please add some information about it and describe observed secondary effect (if any) on goats |
We have added one of the side effects but we have not observed this effect in our work. |
|
Line 269: Did you analyze the minerals of diets (Na for example)? |
Unfortunately, we did not evaluate the mineral analysis. |
|
Line 274 - 275: check English, or rewrite, and explain |
We have rewritten. |
|
Line 278: see previous note about methanol in the diet |
We have added one of the side effects but we have not observed this effect in our work. |
|
Line 290-292: not clear; rewrite |
We have rewritten the sentence. |